# Beyond Platinum, ICIs in Metastatic Cervical Cancer: A Systematic Review

**DOI:** 10.3390/cancers14235955

**Published:** 2022-12-01

**Authors:** Brigida Anna Maiorano, Mauro Francesco Pio Maiorano, Davide Ciardiello, Annamaria Maglione, Michele Orditura, Domenica Lorusso, Evaristo Maiello

**Affiliations:** 1Oncology Unit, Fondazione Casa Sollievo della Sofferenza IRCCS, 71013 San Giovanni Rotondo, Italy; 2Department of Translational Medicine and Surgery, Catholic University of the Sacred Heart, 00168 Rome, Italy; 3Obstetrics and Gynecology Unit, Department of Biomedical Sciences and Human Oncology, University of Bari “Aldo Moro”, 70121 Bari, Italy; 4Medical Oncology, Department of Precision Medicine, Luigi Vanvitelli University of Campania, 80131 Naples, Italy; 5Obstetrics and Gynecology Department, Fondazione Casa Sollievo della Sofferenza IRCCS, 71013 San Giovanni Rotondo, Italy; 6Department of Women and Child Health, Division of Gynaecologic Oncology, Fondazione Policlinico Universitario “A. Gemelli” IRCCS, 00168 Rome, Italy; 7Scientific Directorate, Fondazione Policlinico Universitario “A. Gemelli” IRCCS, 00168 Rome, Italy

**Keywords:** cervical cancer, HPV, PD-L1, ICI, checkpoint inhibitor, immunotherapy, pembrolizumab, nivolumab, ipilimumab, cemiplimab

## Abstract

**Simple Summary:**

Approaches beyond first-line chemotherapy to treat advanced cervical cancer (CC) are currently limited. Immune checkpoint inhibitors (ICIs) are showing high efficacy, thus remodeling the therapeutic scenario of many solid tumors. With our systematic review, we aimed to summarize the latest clinical trials using ICIs in CC. Our systematic review managed to demonstrate that ICIs might represent an appealing strategy for advanced CC, with 2 out of 3 patients responding to ICIs without further concerns about safety. PD-L1 status might be an indicator of response; however, the search for new predictive biomarkers is mandatory. Further studies are needed for appropriate patient selection and a tailored approach.

**Abstract:**

Background: Cervical cancer (CC) constitutes the fourth most common tumor among the female population. Therapeutic approaches to advanced CC are limited, with dismal results in terms of survival, mainly after progression to platinum-based regimens. Immune checkpoint inhibitors (ICIs) are remodeling the therapeutic scenario of many solid tumors. The role of ICIs in CC should be addressed. Therefore, we systematically reviewed the latest clinical trials employing ICIs in advanced CC to assess which ICIs have been employed and how ICIs might meet the need for new therapeutic options in terms of efficacy and safety. Methods: The review was conducted following the PRISMA guidelines. The following efficacy outcomes were specifically collected: overall response rate (ORR), disease control rate (DCR), progression-free survival (PFS), and overall survival (OS); for safety: type, number, and grade of adverse events (AEs). Results: A total of 17 studies were analyzed. Anti-PD1 (pembrolizumab, nivolumab, cemiplimab, balstilimab, and tislelizumab), anti-PD-L1 (atezolizumab), and anti-CTLA-4 (ipilimumab, zalifrelimab) agents were employed both as single agents or combinations. Overall ORR ranged from 0% to 65.9%. ORR ranged from 5.9% to 69.6% in PD-L1-positive patients and from 0% to 50% in PD-L1-negative patients. DCR was 30.6–94.1%. mPFS ranged from 2 to 10.4 months. mOS ranged from 8 months to not reached. PD-L1 status did not impact survival. A total of 33.9% to 100% of patients experienced AEs. Conclusion: Immunotherapy represents an appealing strategy for patients with advanced CC, as 2 out of 3 patients seem to respond to ICIs. PD-L1 status might be an indicator of response without impacting survival.

## 1. Introduction

With an incidence of 15.6 per 100,000 inhabitants per year, cervical cancer (CC) represents the fourth most common cancer among the female population, as well as the fourth cause of cancer-related death worldwide, bearing a mortality rate of 8.8 deaths per 100.000 inhabitants per year [1]. The median age at diagnosis is 49. A dual peak of incidence of CC is registered among the 40–64 and the 65–74 age subgroups, respectively, with 1.8 and 2.4 cases per 100,000 inhabitants per year [1,2]. The 5-year relative survival is 66.3%, as CC is often diagnosed at an early stage on account of early human papillomavirus (HPV) infection detection, thus presenting localized in 44% of cases or spread to regional lymph nodes in 36% of cases. However, 16% of CC cases are diagnosed at the metastatic stage, with a 5-year relative survival dropping to only 17.6% [3].

### 1.1. Treatment Options in Advanced CC

Surgery or definitive radiotherapy are considered the primary treatments for early-stage disease, while concurrent platinum-based chemotherapy and radiotherapy (CTRT) represent the standard of care in the locally advanced disease setting. RT or CTRT is also feasible for recurrences after surgery without previous adjuvant RT. Instead, pelvic exenteration remains the only therapeutic option for women with central pelvic recurrence after RT. Pharmacological approaches to patients with distant or loco-regional recurrences, not eligible for surgery or RT, are currently limited [4,5]. 

In the metastatic setting, platinum-based chemotherapy plus bevacizumab is used as the first choice, with a median overall survival (mOS) of 17 months. However, therapeutic options after progression to first-line therapy are limited, and survival is dismal in this stage, with less than one year of OS [6,7,8,9]. Thus, the search for new therapeutic approaches is an unmet need for advanced CC. 

### 1.2. HPV Infection in CC

Persistent HPV infection is commonly known as the cause of nearly all CC cases, with HPV-16 alone responsible for over 50% of all CCs globally, particularly among the Caucasian population [10,11,12,13]. Despite progress in early HPV detection and extensive vaccination programs, CC still holds one of the highest burdens of disease globally, notably in low-income countries, thus having a significant impact on women’s health worldwide [1,2,3]. HPV infection determines the production of E6 and E7 proteins, with an inhibitory role for the onco-suppressors p53 and Rb [14]. It has been widely demonstrated that HPV infection is responsible for a specific immune response, as an HPV16 E2- and E6-targeted T-helper immune response has been shown in healthy subjects, which might be crucial for controlling HPV infections. HPV can boost the immune response, recruiting E6- and E7-specific T cells, but this mechanism seems lacking in CC patients. Hence, an impaired CD4^+^ T-cell immunity against E2 and E6 antigens has been seen among CC patients, mainly lacking Interferon (IFN)-gamma and Interleukin (IL)-5 production, if compared with healthy subjects [10]. Activating the immune response against the viral infection represents an attractive approach for therapies targeting the immune system, such as immune checkpoint inhibitors (ICIs).

### 1.3. ICIs and PD-L1 in CC and Aims of the Systematic Review

During the last 10 years, ICIs have modified the therapeutic landscape of many solid tumors, and their application in gynecological malignancies has been intensively investigated [15,16]. Removing the brake pedal by inhibiting negative immune checkpoints such as Programmed Death 1 (PD1), PD-Ligand 1 (PD-L1), and Cytotoxic T-lymphocyte-associated protein 4 (CTLA4), ICIs could produce a robust antitumor activity [15,16,17]. 

Compared to endometrial and ovarian cancer, CC has a higher rate of PD-L1, as up to 80% of squamous and around 65% of adenocarcinomas are PD-L1-positive, and CD8^+^ T cells express PD1 [18,19]. However, while the increased expression of PD-L1 has been associated with poorer prognosis or lower OS in other tumor subtypes, this is not the case for CC [20]. PD-L1 has already been addressed as potentially associated with a better ICIs response in CC patients. Additionally, other factors could justify a suitable response to ICIs: a high tumor mutational burden (TMB) and, therefore, a high neo-antigens load that can stimulate immune activation; also, around 8–10% of CC carry a deficit of mismatch-repair genes leading to microsatellite instability (MSI) [21]. 

ICIs trials have also been conducted in CC. However, there is currently a gap in knowledge regarding the role of ICIs in the treatment scenario of advanced CC patients. We hereby systematically reviewed the latest clinical trials regarding the use of ICIs for CC treatment to address which agents have been employed and assess how ICIs might meet the need for new therapeutic options, notably in the advanced or recurrent CC setting, in terms of response rate and survival, and, finally, the safety profile. To our knowledge, this is the first systematic review analyzing the use of ICIs in the advanced CC setting.

## 2. Materials and Methods

### 2.1. Protocol Registration

We registered the protocol for this systematic review with PROSPERO (CRD42022314512).

### 2.2. Search Strategy and Data Extraction

This systematic review was carried out following the Preferred Reporting Items for Systematic Reviews and Meta-Analysis (PRISMA) statements [22]. Two authors (MFPM and BAM) independently performed a literature search of the databases PubMed, EMBASE, and Cochrane Central Register of Controlled Trials, on March 31, 2022. The search terms (“cervical neoplasms” OR (“cervical” AND “neoplasms”) OR “cervical cancer” OR (“cervical” AND “cancer”) OR “cervix cancer” OR (“cervix” AND “cancer”) AND (‘’immune checkpoint inhibitors” OR “ICIs” OR “avelumab” OR “nivolumab” OR “atezolizumab” OR “pembrolizumab” OR “durvalumab” OR “cemiplimab” OR “tremelimumab” OR “ipilimumab” or “dostarlimab” OR “balstilimab” OR “camrelizumab”) were used. An additional search for conference abstracts from the American Association of Clinical Oncology (ASCO), European Society of Medical Oncology (ESMO), Society of Gynecologic Oncology (SGO) was performed. Article citations were manually checked for additional references.

### 2.3. Inclusion and Exclusion Criteria, Population, Intervention, and Outcomes

We included phase I-IV clinical trials reporting efficacy and safety data of ICIs (single agents or combinations) in metastatic/recurrent CC patients, written in the English language. From multicohort trials, the number and data of CC patients were identified. Meta-analyses, reviews, case reports, correspondences, personal opinions, and in vitro/animal studies were excluded. For the selected studies, the following data were collected: trial name, first author, year of publication, phase, number of treated patients, administered drugs and dosage, and primary and secondary endpoints. We specifically addressed the following efficacy outcomes: overall response rate (ORR), disease control rate (DCR), progression-free survival (PFS), and overall survival (OS) for safety, number, and grade of treatment-related adverse events (AEs) (Appendix A).

### 2.4. Risk of Bias

Two reviewers independently assessed the risk of bias. In case of disagreement, a third reviewer would have been consulted. The Risk Of Bias In Non-randomized Studies of Interventions (ROBINS-I) tool was used to assess the risk of bias, including eight domains: confounding bias; selection bias; classification bias; deviation from intended interventions bias; missing data; measure outcome bias; selection of the reported results; overall bias [23].

## 3. Results

A total of 124 studies were identified via electronic research. A total of 116 studies were eligible after duplicate removal and screening based solely on title and abstract analyses. A total of 6 studies were written in languages other than English; 30 case reports, reviews, correspondences, personal opinions, or commentaries were removed; the complete text was not available in 1 study, while 31 were removed for focusing on different topics after applying inclusion and exclusion criteria. Therefore, a total of 17 studies were included in our review (Figure 1).

### 3.1. Characteristics of the Included Studies

The included studies were 2 phase I, 2 phase I/II, 11 phase II clinical trials, and 2 randomized phase III trials [24,25,26,27,28,29,30,31,32,33,34,35,36,37,38,39,40,41]. Anti-PD1 agents were used in 14 studies (pembrolizumab in 4 studies; nivolumab in 3 studies; cemiplimab, balstilimab, and camrelizumab in 2 studies each, tislelizumab in 1 study), 2 studies employed the anti-PD-L1 atezolizumab. Anti-PD1 drugs were administered as single agents in 8 studies [24,26,28,29,31,32,34,35]. Nivolumab was combined with ipilimumab in two cohorts of the CheckMate 358 study [30]. In one study, the anti-CTLA-4 ipilimumab was administered as a single agent [41]. Balstilimab was combined with the anti-CTLA4 zalifrelimab in one study [33]. Pembrolizumab was combined with a DNA vaccine in 1 study, and chemotherapy plus bevacizumab in one study [25,27]. In three studies, ICIs were combined with anti-angiogenics: camrelizumab plus the tyrosine-kinase inhibitor (TKI) apatinib, tislelizumab plus the TKI anlotinib, and atezolizumab plus bevacizumab [36,38,39]. 

Overall, a total of 2114 patients were treated, ranging from 11 to 617. ORR-defined as the percentage of patients achieving a complete response (CR) or a partial response (PR)-was the most frequent primary endpoint (13 studies) [24,25,26,28,29,30,31,32,33,36,37,38,39,41]. Safety was the primary endpoint in two studies [35,41]. PFS-defined as the time from randomization to disease progression or death, whichever occurred first, and OS-defined as the time from randomization to death-were assessed as co-primary endpoints in one case [27]. In another study, OS alone was the primary endpoint [34]. Non-progression rate (NPR)-defined as the percentage of CR + PR + stable disease (SD) at 18 weeks-was the primary endpoint in 1 study [40]. DCR–defined as the percentage of patients achieving a CR/PR or SD, PFS, OS, duration of response (DoR), and safety were investigated among secondary endpoints. The studies were all conducted among pre-treated patients, except from 1 study of camrelizumab plus chemotherapy, and a group of naive patients in the Keynote-826 [27,37]. 

Table 1 resumes the main characteristics of the included studies.

Overall, ORR ranged from 0% to 65.9%, reaching 26.3% with single-agent ICIs, 65.9% in combination studies with chemotherapy, and 55.6% with TKIs. In PD-L1-positive patients, ORR ranged from 5.9% to 33% with single-agent ICIs, from 27% to 36% with dual ICIs, reaching 68.1% and 69% when ICIs were combined with chemotherapy and TKIs, respectively. In PD-L1-negative women, ORR was 0–16.7% to single-agent ICIs, 11–35.8% to double ICIs, and reached 50% after ICIs plus TKIs (Figure 2). 

A total of 8 studies reported DCR, ranging from 30.6% to 94.1%. 

A total of 13 studies reported mPFS that ranged from 2 to 10.4 months. mOS was reported by 14 studies and ranged from 8 months to NR (Figure 3). 

A total of 33.9% to 100% of patients developed AEs, of which up to 81.8% were over grade 3 (≥G3). G3 AEs were identified as severe or medically significant but not immediately life-threatening AEs, according to the Common Terminology Criteria for Adverse Events (CTCAE) definition [42]. In combination, a higher toxicity rate was reported. 

Table 2 resumes the main results of the included studies. 

No serious risk of bias emerged (Appendix A).

### 3.2. ICIs Targeting PD1

#### 3.2.1. Pembrolizumab

Pembrolizumab binds to the PD-1 receptor, blocking immune-suppressing ligands PD-L1 and PD-L2 from interacting with PD-1, and represents one of the most employed ICIs in the daily clinical practice of many solid tumors. Pembrolizumab plays a key role in CC, as after KEYNOTE-158 and KEYNOTE-826 it was approved for PD-L1-positive CC patients as a single agent after chemotherapy progression, and plus chemotherapy in first line [43,44,45]. A total of 4 studies using pembrolizumab were included in our systematic review [24,25,26,27]. Patients were not selected for PD-L1, or HPV, except from one study that included only HPV 16/18^+^ patients [25].

##### Pembrolizumab as Single Agent

In the CC cohort of the phase Ib KEYNOTE-028 trial (NCT02054806), 24 women with PD-L1-positive (≥1%), advanced, pre-treated CC received pembrolizumab 10 mg/kg q2w for up to 24 months. ORR by RECIST was the primary endpoint, safety was the secondary endpoint. After a median follow-up of 11 months, ORR was 17% (95% confidence interval [CI], 5–37%), with 4 PRs and 3 SDs. mDOR was 5.4 months, ranging from 4.1 to 7.5 months. Treatment-related AEs (TRAEs) were reported in 75% of patients, but no G4 AEs or treatment-related deaths were reported. mPFS was 2 months (95% CI, 2–3 months), mOS was 11 months (95% CI, 4–15 months) [24]. 

The phase II KEYNOTE-158 study (NCT02628067) evaluated the efficacy and safety of pembrolizumab in solid tumors, including previously treated advanced CC. A total of 98 women received pembrolizumab 200 mg q3w (for a maximum of 2 years). ORR per RECIST was the primary endpoint, safety was the secondary endpoint. A total of 83.7% of patients had PD-L1-positive tumors, defined as combined positive score (CPS)≥1. ORR was 12.2% (95% CI, 6.5–20.4%), and DCR 30.6% (95% CI, 21.7–40.7%), with 3 CRs, 9 PRs, 18 SDs. All 12 responses and 15/18 SDs were found in the PD-L1-positive group, with ORR 14.6% (95% CI, 7.8–24.2%) and DCR 32.9% (95% CI, 22.9–44.2%). Responses seemed to be durable, with an mDoR not reached (NR) after a median follow-up of 10.2 months. mPFS was 2.1 months (95% CI, 2.0–2.2 months), mOS was 9.4 months (95% CI, 7.7–13.1 months) in the total population and 11 months (95% CI, 9.1–14.1 months) in the PD-L1-positive subgroup. A total of 65.3% of patients experienced TRAEs, including 12.2% of patients that experienced ≥G3 events, more frequently an increasing in transaminase levels. No treatment-related deaths were recorded [26]. Based on the results of this study, in 2018, the Food and Drug Administration (FDA) approved pembrolizumab for pre-treated CC patients with CPS ≥ 1 for PD-L1 [42]. 

##### Pembrolizumab Combinations

In the single-arm, phase II NCT03444376 trial, 36 patients with advanced, pre-treated HPV-16- or -18-positive CC were recruited to receive pembrolizumab 200 mg q3w and the DNA vaccine GX-188E. Patients were tested with HPV tests, such as Roche^®^ HPV Test (Roche Diagnostics; Basel, Switzerland), Qiagen^®^ Hybrid Capture 2, or Seegene^®^ Anyplex II HPV HR Detection kit, with archival or fresh biopsy samples. ORR was the primary endpoint, while safety, DoR, OS, and 6-months PFS were secondary endpoints. A total of 26 patients were eligible for antitumor activity analyses. An overall ORR of 42% (95% CI, 23–63%) was reached, with 4 CRs (15%), 7 PRs (27%), 4 SDs (15%) and 11 progressive diseases (PDs-41%), and a DCR of 58% (95% CI, 37–77%). ORR was 50% (95% CI, 27–73%) and DCR 65% (95% CI, 41–85%) in the PD-L1-positive subgroup, while ORR was 17% (95% CI, 0–64%) and DCR 29% (95% CI, 4–78%) in the PD-L1-negative one. In the HPV-16-positive group, ORR was 45% (95% CI, 23–68%) and DCR 60% (95% CI, 36–8%); in the HPV-18-positive group, ORR was 33% (95% CI, 4–78%) and DCR 50% (95% CI, 12–88%). Of note, all four observed CRs were PD-L1 and HPV16-positive squamous CCs. DoR ranged from 3.3 to 13.6 months. mPFS was 4.9 months (95% CI, 2.1–6.7 months), with 6-month PFS 35%. mOS was 10.2 months (95% CI, 6.6–16.7 months). TRAEs were experienced by 16 patients (44%), with gastrointestinal problems being the most common reported G1/2 AEs. There were also 4 (11%) ≥G3 AEs, with one pericardial effusion being the most serious one [25]. The efficacy and safety of pembrolizumab plus GX-188E were recently updated at ESMO 2022: among 60 evaluated patients (36 PD-L1-positive and 24 PD-L1-negative), 6 CRs and 13 PRs were found, reaching an ORR of 31.7%, notably showing significant efficacy also in the PD-L1-negative subgroup, with an ORR of 25%. mDOR and mOS were 12.3 and 17.2 months, respectively. TRAEs were reported in 33.8% of patients (22/65), with 3 G3/4 AEs (4.6%) [46].

A total of 617 women with not pre-treated advanced CC were randomized 1:1 in the phase III double-blind KEYNOTE-826 trial (NCT03635567) to receive pembrolizumab 200 mg or placebo (PBO) q3w for up to 35 cycles, added to paclitaxel (175 mg/m2) and investigator’s choice between cisplatin (50 mg/m2) or carboplatin (5 mg/mL/min). Patients could also receive bevacizumab 15 mg/kg q3w. OS and PFS by RECIST were co-primary endpoints, while DoR, ORR, and PFS rate at 12 months were secondary endpoints. Three populations were analyzed: PD-L1-positive patients with CPS ≥ 10, PD-L1-positive with CPS 1-10, and all comers. The first interim analysis was pre-planned in the PD-L1-positive patients (defined as CPS ≥ 1). A total of 88.6% of patients in the pembrolizumab group and 89% in the PBO group had PD-L1-positive cancers. Bevacizumab was administered to 63.6% of patients in the pembrolizumab arm and 62.5% in the PBO arm. The results showed that PFS was significantly longer in the pembrolizumab arm compared to PBO, achieving an mPFS of 10.4 months (95% CI, 9.1–12.1 months) versus 8.2 months (95% CI, 6.4–8.4 months–hazard ratio (HR) 0.65; 95% CI, 0.53–0.79, *p* < 0.001), in the intention-to-treat (ITT) population. Results were particularly interesting in the PD-L1-positive subgroup, reaching PFS of 10.4 months (95% CI, 9.7–12.3 months) versus 8.2 months (95% CI, 6.3–8.5 months-HR 0.62; 95% CI, 0.50–0.77, *p* < 0.001) with CPS ≥ 1, and 10.4 months (95% CI, 8.9–15.1 months) versus 8.1 months (95% CI, 6.2–8.8 months-HR 0.58; 95% CI, 0.44–0.77; *p* < 0.001) with CPS ≥ 10. mOS was not reached in both arms; however, the 24-month estimate of patients alive favored the pembrolizumab arm with a percentage of 53.0% vs. 41.7% (HR 0.64; 95% CI, 0.50–0.81; *p* < 0.001), 50.4% vs. 40.4% (HR, 0.67; 95% CI, 0.54–0.84; *p* < 0.001), and 54.4% vs. 44.6% (HR, 0.61; 95% CI, 0.44 to 0.84; *p* = 0.001), for the PD-L1 CPS ≥ 1, the ITT and CPS ≥ 10 groups, respectively. Higher rates of confirmed responses were reported in the pembrolizumab arm for all groups (65.9% vs. 50.8%, 68.1% vs. 50.2%, and 69.6% vs. 49.1%, respectively). A total of 14 possible treatment-related deaths were reported in both arms; in the pembrolizumab group, a slightly higher number of ≥G3 TRAEs (81.8% vs. 75.1%) and more immune-related AEs (irAEs-33.9% vs. 15.2%) were observed, with one death potentially due to an irAE [27]. 

In conclusion, these data demonstrate that pembrolizumab has efficacy in CC patients, with the maximum advantage in the case of PD-L1 positivity, without particular safety concerns. At the moment, the use in pre-treated patients as monotherapy, and combined with chemotherapy in naïve patients, is justified by these results, which led to the approvals by the FDA in October 2021, and the European Medical Agency (EMA) in April 2022, of pembrolizumab plus chemotherapy combination for the first-line treatment, and pembrolizumab monotherapy for pre-treated patients, in case of CC with PD-L1 positivity defined as CPS ≥ 1 [43,44]. Only one study specifically focused on HPV^+^ patients, which seemed to achieve reasonable disease control with ICIs, especially when HPV and PD-L1 positivity are detected together. 

#### 3.2.2. Nivolumab

Nivolumab, another historical anti-PD1 agent, was used as monotherapy in pre-treated patients. Of note, even if most patients were PD-L1-positive, PD-L1 status was not an inclusion criterion for the selected studies [28,29,31]. We also reviewed the studies of nivolumab combinations, as the association with the anti-CTLA4 ipilimumab was the first double ICIs combination used in CC patients [30].

##### Nivolumab as Single Agent

Nivolumab was tested as a single agent at the flat dose of 240 mg q2w in one phase I/II study and 3 mg/kg q2w in one phase II trial. 

In the phase II NCT02257528/NRG-GY002 trial, patients with advanced, pre-treated CC received nivolumab 3 mg/kg q2w until progression or unacceptable toxicity. The primary endpoints were ORR by RECIST. A total of 17 patients were PD-L1-positive (CPS ≥ 1%). Among 25 treated patients, ORR was 4% (90% CI, 0.4−22.9%), with a DoR of 3.8 months. Of note, the only confirmed PR and 7 out 9 SDs were observed in the PD-L1-positive subgroup, thus reaching an ORR of 5.9%. After a median follow-up of 32 months, mPFS was 3.5 months (90% CI, 1.9–5.1 mos) and mOS 14.5 months (90% CI, 8.3–26.8 mos). A total of 84% of patients experienced TRAEs, with 32% ≥G3 events, one patient discontinuing nivolumab due to hepatic toxicity and two patients experiencing G4 increase in serum amylase and bilirubin levels [28]. 

In the JapicCTI-163212 phase II trial, nivolumab was administered to Japanese patients with advanced CC, uterine cancer, and soft tissue sarcomas. In the CC cohort, 20 pre-treated women were treated. ORR, the primary endpoint, was 25% (95% CI, 13–41%), ranging from 0% in PD-L1-negative patients (*n* = 5) to 33% in PD-L1-positive patients (*n* = 15; CPS ≥ 1). Secondary endpoints included PFS, OS, DCR, and DoR. mPFS was 5.6 months (95% CI, 2.8–7.1 mos); mOS and mDoR were NR with 6-mos OS 84% (95% CI, 70–92%). A total of 65% of patients developed TRAEs, with 20% ≥G3 [31].

The phase I/II multicohort CheckMate 358 trial (NCT02488759) evaluated the use of nivolumab 240 mg q2w in several HPV-related tumors, including 19 patients with advanced refractory CC. The primary endpoint was ORR. A total of 10 patients were PD-L1^+^ (62.5%; CPS ≥ 1). Overall, ORR was 26.3% (95% CI, 9.1–51.2%), and DCR was 68.4% (95% CI, 43.4–87.4%). Among PD-L1^+^ patients, ORR was 20% (95% CI, 2.5–55.6%), and DCR 70% (95% CI, 34.8–93.3%). In PD-L1^−^ patients, ORR was 16.7% (95% CI, 0.4–64.1%), and DCR 50% (95% CI, 11.8–88.2%). mDOR was NR. mPFS was 5.1 months (95% CI, 1.9 to 9.1 mos), while 21.9 months (95% CI, 15.1 mos-NR) was the mOS. A total of 63.2% of patients experienced at least one TRAE of any grade, with 21.1% reporting ≥G3 TRAEs, most commonly diarrhea and fatigue [29].

##### Nivolumab + Ipilimumab

The CheckMate 358 study was recently updated with new results, as 176 patients with advanced CC were treated, receiving nivolumab 240 mg q2w (NIVO, *n* = 19), nivolumab 3 mg/kg q2w plus ipilimumab 1 mg/kg q6w (N3I1, *n* = 45) or nivolumab 1 mg/kg plus ipilimumab 3 mg/kg q4w for 4 cycles followed by nivolumab 240 mg q2w (N1I3, *n* = 112). ORR was 26.3%, 31.1%, and 38.4% for the NIVO, N3I1, and N1I3 groups, respectively, and responses were observed regardless of PD-L1 status. Among PD-L1 patients, ORR was 27.3%, 36%, and 35.8%, while among PD-L1-negative patients was 14.3%, 20%, and 30.6%, respectively. No new safety concerns emerged, except for hepatitis, reported in 16% of patients in the N1I3 subgroup [30].

Together, these data show that nivolumab is a reliable option in pre-treated CC patients, with higher efficacy in the case of PD-L1 positivity. Differently, no prognostic role of PD-L1 is evidenced with the combination of nivolumab and ipilimumab. 

#### 3.2.3. Balstilimab

Among newer anti-PD1 agents, balstilimab has been employed both as monotherapy and in combination with another anti-CTLA4 drug, zalifrelimab [32,33]. 

A total of 161 women with advanced pre-treated CC were enrolled to receive balstilimab 3 mg/kg q2w (24 months maximum) in the NCT03104699 phase II trial, whose primary endpoint was ORR, while DoR and DCR were secondary endpoints. A total of 140 patients were included for efficacy analyses: 99 of them (61.5%) had PD-L1^+^ tumors (CPS ≥ 1%), and 43 (26.7%) were PD-L1-negative. With 5 CRs (3.6%), 16 PRs (11.4%), 51 SDs (36.4%), ORR was 15% (95% CI, 10–21.8%), and DCR of 49.3% (95% CI, 41.1–57.5%). Responses were durable, with an mDOR of 15.4 months (95% CI, 5.7 months-NR). ORR among PD-L1^+^ patients was 20.0% (95% CI, 12.9–29.7%) and 7.9% among PD-L1-negative women. Results were independent of histology or previous treatment with bevacizumab. TRAEs were experienced by 71.4% of patients, the most common being asthenia (23%) and diarrhea (12.4%). ≥G3 TRAEs occurred in 11.8% of patients [32]. 

In the NCT03495882 phase II trial, 143 patients with pre-treated advanced CC received the combination of balstilimab 3 mg/kg q2w with zalifrelimab 1 mg/kg q6w (up to 2 years). A total of 55% of patients were PD-L1-positive (defined as CPS ≥ 1), 25% PD-L1-negative. ORR-the primary endpoint was 22%, ranging from 11% among PD-L1-negative patients to 27% among PD-L1-positive patients. mDOR was NR. A total of 35% irAES were detected, of whom 10.5% were ≥G3. A total of 10% of patients discontinued the treatment, and two deaths were recorded [33].

These data show that balstilimab is a newer ICI that could be further investigated in CC patients. Similarly to previous trials, there is a slight tendency for higher responses in PD-L1-positive than PD-L1-negative patients, also in the combination studies.

#### 3.2.4. Cemiplimab

We also searched for studies using the anti-PD1 agent cemiplimab, which is being investigated in many tumor subtypes with satisfactory efficacy [34,35]. Cemiplimab was tested only as monotherapy in two studies, a phase I (3 mg/kg) and a phase III (flat dose 350 mg q3w). EMPOWER-Cervical 1/GOG-3016/ENGOT-cx9 (NCT03257267) was a phase III clinical trial evaluating cemiplimab 350 mg q3w versus investigator’s choice single-agent chemotherapy in 608 patients. The study met its primary endpoint, as mOS was 12 months with cemiplimab versus 8.5 months with chemotherapy (12.0 vs. 8.5 mos-HR 0.69; 95% CI, 0.56–0.84; *p* < 0.001). The cemiplimab group also reached a longer PFS (HR 0.75; 95% CI, 0.63–0.89; *p* < 0.001). ORR was 16.4% (95% CI, 12.5–21.1%) in the cemiplimab versus 6.3% (95% CI, 3.8–9.6%) in the chemotherapy subgroup. Patients were enrolled independently from PD-L1 expression; however, subgroup analysis suggested a predictive role of PD-L1. In fact, ORR to cemiplimab was higher in PD-L1-positive (18.3%; 95% CI 10.6–28.4%; TPS ≥ 1%) than in negative patients (11.4%; 95% CI, 3.8–24.6%), as well as mOS (13.9 vs. 7.7 months). Cemiplimab was also better tolerated, as 45% of patients experienced ≥G3 AEs, compared to 53.4% with chemotherapy [34]. Cemiplimab also showed to prolong survival despite CC histotypes, as mOS was 10.9 vs. 8.8 months in the squamous cell carcinoma subgroup and 13.5 vs. 7.0 months in the adeno- or adenosquamous subgroup [47].

In the expansion cohorts 23 and 24 of a phase I trial (NCT02760498), 20 patients with advanced pre-treated CC were treated with cemiplimab 3 mg/kg q2w for 48 weeks of monotherapy (*n* = 10) or in combination with hypo-fractionated RT (*n* = 10), regardless of PD-L1 expression and histology. Safety was the primary endpoint; ORR, DCR, DoR, PFS, and OS were secondary endpoints. ORR was 10% (95% CI, 0.3–44.5%) for both cohorts, as 1 patient for each cohort achieved a PR, while DCR was 40% (95% CI, 12.2–73.8%) in the monotherapy cohort and 60% (95% CI, 26.2–87.8%) in the association cohort. DoR was 11.2 months vs. 6.4 months, respectively. mPFS was 1.9 months (95% CI, 1–9 mos) and mOS 10.3 months (95% CI, 2.1-NE) for the monotherapy group; in the association group, mPFS was 3.6 months (95% CI, 0.6–5.7 mos) and mOS 8 months (95% CI, 1.7 mos-NR). TRAEs were experienced by 90% and 100% of patients in the monotherapy and association groups, with ≥G3 TRAEs experienced by 10% and 30% of patients, respectively [35].

The results of these trials confirm that cemiplimab could be used in pre-treated CC patients, also with rare histologic subtypes, with a potential predictive role of PD-L1 for cemiplimab versus chemotherapy. No safety concerns, but dismal efficacy emerged from the combination of cemiplimab with RT. 

#### 3.2.5. Camrelizumab

In a single-arm, phase II trial (NCT03816553), 45 patients with advanced, pre-treated CC were given camrelizumab 200 mg q2w and the tyrosine-kinase inhibitor (TKI) apatinib. The primary endpoint was ORR, and the secondary endpoints were PFS, OS, DoR, and safety. A total of 41 patients were included in the efficacy analyses. A total of 66.7% of patients were PD-L1^+^ (CPS ≥ 1). In the ITT population, ORR was 55.6% (95% CI, 40–70.4%), with 2 CRs and 23 PRs. mPFS was 8.8 months (95% CI, 5.6 mos-NR), mOS and mDOR were NR. Post hoc analyses showed that ORR was 69% for PD-L1-positive and 50% for PD-L1-negative patients. The PD-L1-positive subgroup also achieved a longer PFS: NR (95% CI, 5.8 mos-NR) vs. 5.2 months (95% CI, 1.6 mos-NR) of PD-L1-negative. A total of 95.6% of patients experienced at least one TRAE, and 71.1% experienced ≥G3 TRAEs, with hypertension (24.4%), anemia (20%), and fatigue (15.6%) being the most common. A total of 15 patients (33.3%) experienced irAEs [36]. 

In another phase II prospective study, camrelizumab 200 mg was given as first-line treatment in combination with albumin-binding paclitaxel 260 mg/mq and carboplatin area under the curve (AUC) 5 q3w for 6 cycles, followed by camrelizumab 200 mg maintenance q3w, to 35 patients with advanced CC. A total of 27 patients were eligible for efficacy analyses. The ORR was 40% (95% CI, 21.13–61.33%), and the DCR was 92% (95% CI, 73.97–99.01%). A total of 4 CRs, 6 PRs, and 13 SDs were observed. The most common AE was reactive cutaneous capillary endothelial proliferation (RCCEP), reported in 23 (65.71%) patients. G3 AEs included 5 (14.29%) myelosuppression, and G4 AEs included 1 (2.86%) RCCEP and 1 (2.86%) bladder inflammation [37].

In conclusion, these data constitute a rationale for combining ICIs with agents having different mechanisms of action, such as TKIs and chemotherapy.

#### 3.2.6. Tislelizumab

The anti-PD1 tislelizumab 200 mg was administered with anlotinib 10 mg qd for 14 days q3w in a phase II trial. A total of 32 patients were enrolled, while 17 were evaluated as they received at least four cycles of treatment. An ORR of 35.3% emerged (95% CI, 17.3–58.7%), and DCR was 94.1% (95% CI, 73–98.9%). mPFS and OS were NR. All patients experienced G1/2 TRAEs, while only 0.06% experienced G3 TRAEs [38]. The combination of these two agents seems, therefore, effective and safe for CC patients.

### 3.3. ICIs Targeting PD-L1

We subsequently searched for trials using anti-PD-L1 agents in CC patients; only atezolizumab studies were found [39,40].

The NCT02458638 phase II trial used atezolizumab (1200 mg q3w) monotherapy in 16 cohorts with different advanced solid tumors (*n* = 433), CC included. Among 27 CC patients, ORR was 14.8%, with 1 CR and 3 PRs; DoR ranged from 2.99 months to 1.27 years. The mPFS was 4.14 months (95% CI, 1.31–8.34 mos), while the mOS 14.78 months (95% CI, 10.55–26.51 mos). TRAEs for the CC cohort were experienced by 64.3% of patients, with only 10.7% ≥G3 [40].

The NCT02921269 phase II study employed the anti-PD-L1 agent atezolizumab (1200 mg q3w). Combined with bevacizumab, atezolizumab was given to 11 women with advanced, pre-treated CC, with ORR by RECIST as the primary endpoint and DCR, PFS, and OS as secondary endpoints. The study did not meet its primary endpoint, as zero patients achieved an objective response (ORR 0%). DCR was 60% (6 SDs). mPFS was 2.9 months (95% CI, 1.8–6 mos), and mOS 8.9 months (95% CI, 3.4–21.9 mos). TRAEs were experienced by 72% of patients and ≥G3 AEs by 36.4%. Two high-grade neurologic events were reported [39]. 

These data do not seem to support the use of anti-PD-L1 agents in combination with bevacizumab in CC patients.

### 3.4. ICIs Targeting CTLA-4

As in other tumor subtypes, we reviewed the inhibition of the CTLA4 pathway with single agents and found a unique trial with ipilimumab. As monotherapy, ipilimumab was administered to 42 patients in a phase I/II NCT01693783 study at the dosage of 10 mg/kg q3w for four doses, followed by maintenance with four cycles q12w. A total of 1 PR and 10 SDs were recorded, thus resulting in an ORR of 2.9%. mPFS was 2.5 months (95% CI, 2.1–3.2 mos), and mOS was 8.5 months (95% CI, 3.6 mos-NR). Results were not influenced by PD-L1 expression. G3 TRAEs were reported in four patients (9.5%), with three having severe colitis [41]. Therefore, the inhibition of CTLA-4 with single-agent strategies seems less effective than targeting the PD1/PD-L1 pathway in CC patients.

## 4. Discussion

### 4.1. Summary of Systematic Review Results

Our systematic review confirms the benefit of response and survival in patients diagnosed with advanced CC receiving an ICI. Up to 2 out of 3 patients respond to ICIs. However, results are very heterogeneous due to the designs of the trials, administered agents and combinations, and selection criteria; therefore, a large range, from 0% to 65.9%, of ORR is found. In patients not selected for PD-L1, ORR ranges from 0% to 26.3% with single-agent ICIs, reaching 38.4% with dual ICIs association, 65.9% with the combination of ICIs and chemotherapy, 55.6% with ICIs plus TKIs. In PD-L1-positive patients, ORR ranges from 5.9% to 33% with single-agent ICIs, from 27% to 36% with dual ICIs, reaching 68.1% and 69% when ICIs are combined with chemotherapy and TKIs, respectively. In PD-L1-negative women, ORR is 0–16.7% to single-agent ICIs, 11–35.8% to double ICIs, and reaches 50% after ICIs + TKIs. Therefore, the response rate is driven mainly by PD-L1-positive patients, but PD-L1-negative ones are less represented in the studies, as expected, given the high rate of PD-L1 positivity found in CC.

As already known in other tumor subtypes, ICIs impact survival in advanced CC patients, ranging from 8 to over 21 months, with most studies that have not reached the mOS yet.

The safety profile is manageable with single agents and in line with other ICIs studies in the combination setting.

### 4.2. PD-L1 and Predictive Biomarkers for ICIs Response

KEYNOTE-158 was the only study to include only PD-L1-positive patients [26]. There is no uniformity regarding the method and cut-off used to detect PD-L1 positivity in the different clinical trials (Table 1). Most of them measured the combined positive score (CPS), defined as the number of PD-L1 staining cells divided by the total number of viable tumor cells multiplied by 100. In two studies, the tumor-proportional score (TPS)-defined as the percentage of tumor cells expressing PD-L1, was used. Most studies defined 1% as a cut-off to define PD-L1 positivity. The staining platforms used for PD-L1 detection varied between the studies, with 28–8 (Dako), 22C3, and SP263 (Ventana) antibodies mainly used. Moreover, scores could be at higher risk of inaccuracy when a low cut-off is considered, such as in the CC case.

Although the regulatory approvals of ICIs are based on PD-L1 expression, challenges remain, including variable expression, different antibodies, and staining platforms, and the lack of an unequivocal scoring system, that still now raises the question of whether PD-L1 is the unique suitable biomarker to predict response to ICIs in advanced CC patients. It was previously demonstrated that higher PD-L1 expression and CD8^+^ T cells infiltration predicted a better response to chemo- and radiotherapy, but also ICIs [48,49,50,51,52]. More recently, it has been shown that ICIs administration increased T and B lymphocytes and natural killer (NK) cells infiltration in the tumor microenvironment (TME) in a directly proportional way with ICIs response [53]. A recent analysis found two different clusters of TME could be found in CC patients. In the first cluster, the immune-suppressive TME, a high infiltration of myeloid-derived suppressor cells (MDSCs), macrophages, and Tregs was found. On the other hand, a high infiltration of activated T cells and NK cells was associated with an immune-responsive TME: once again, this subtype seemed to respond better to ICIs [21,53,54]. 

As PD-L1 does not appear as an entirely consistent and reproducible biomarker, other elements with a prognostic and predictive role should be investigated, such as TMB or MS status, which have also been studied in CC, to build a multi-marker classification eventually.

### 4.3. HPV Role in ICIs Response

Notwithstanding the predominant role of HPV in CC development, the trials we reviewed did not attribute a central role to HPV for ICIs response, except for the phase II NCT03444376 trial, which specifically included HPV-16- or -18-positive patients treated with pembrolizumab plus a vaccine [25]. Results were satisfactory in terms of response rate and survival but not particularly different from studies not focusing on HPV-positive patients. As CC is a paradigmatic example of an HPV-dependent neoplasm, vaccines seem feasible and effective, combined with ICIs, to reciprocally potentiate efficacy and overcome resistance. Oncogenes E6 and E7 represent ideal targets for CC therapeutic vaccines [55,56,57,58,59,60,61,62,63,64,65,66]. 

It was previously found that, despite HPV-specific T cells infiltrating the primary and metastatic sites, though not E6- or E7-specific, CC cannot be eradicated, thus suggesting the existence of an immunosuppressive tumor TME [14]. Antigen-presenting cells (APCs) themselves present HPV antigens in a tolerogenic way, activating immune-escaping rather than immune-activating pathways [67,68]. Indeed, the PD-1/PD-L1 axis supports tumor mechanisms for escaping immune response by down-regulating tumor-directed T cells. Therefore, PD-L1 might also be considered a marker of an advancing HPV infection [69]. Even if in a small sample size, the results of NCT03444376 were more satisfactory if HPV-positive patients also expressed PD-L1 [25].

### 4.4. Future Perspectives 

It is of great importance that ICIs efficacy is not counterbalanced by increasing toxicity, also considering the combination setting. Therefore, the upfront combination of chemo- and immunotherapy appears to be a compelling strategy for these patients. New combinations will provide further options for expanding ICIs efficacy and treatment options in CC (Table 3). 

Interesting associations might be those with RT, as radiations induce the production of neo-antigens that boost the immune system and increase CD4^+^ and CD8^+^ infiltration [51,70]. Effectively, in the locally advanced setting, ICIs are being explored in addition to definitive CTRT [71]. Used sequentially after chemoradiation, ipilimumab induced a 1-year PFS of 81% and 1-year OS of 90% in the phase I GOG-9929 trial [72,73]. A total of 52 patients with locally advanced CC were randomized to pembrolizumab (3 doses) after or concurrently with the CTRT regimen in a phase II trial. Safety was the primary endpoint of the analyses: 4 G1 AEs and 23 G3 AEs were reported [74]. However, the phase III CALLA study did not meet its primary endpoint of improving PFS with durvalumab added to CTRT versus CTRT alone [75]. Definitive results of pembrolizumab plus CTRT are expected from ENGOT-cx11/KEYNOTE-A18 in a high-risk CC population [76]. Finally, the triple combination of ICIs, chemotherapy, and anti-angiogenics, as well as the combination of anti-angiogenics and TKIs, have a strong rationale for efficacy and are being explored by several studies [77]. Sequencing strategies after progression to ICIs are needed: combo immunotherapy and novel agents, such as antibody-drug conjugates, could be employed.

### 4.5. Limitations of the Systematic Review

Our systematic review has some limitations. First and foremost, the included trials are heterogeneous in terms of treated patients and biomarkers selection. Moreover, studies are ongoing, with survival data yet to be completed. Furthermore, there is a small number of high-quality studies, such as randomized trials. A quantitative meta-analysis could not be performed, as most of the available trials were not designed in a comparative manner. Therefore, the conclusions about the efficacy and safety of ICIs in CC that can be drawn from our analysis are only descriptive. We are confident that a longer follow-up and a larger number of randomized trials would better clarify the real effect of ICIs on the survival of advanced CC patients.

## 5. Conclusions

Immunotherapy represents an appealing strategy for patients with advanced CC. A subset of patients had a benefit from ICIs with long-lasting responses even in a chemo-refractory setting. Moreover, the combination of chemotherapy and immunotherapy seems to be an effective first-line treatment with acceptable toxicity. Longer follow-ups could confirm these results. Further research is needed for an appropriate patient selection and a tailored approach.

## Figures and Tables

**Figure 1 cancers-14-05955-f001:**
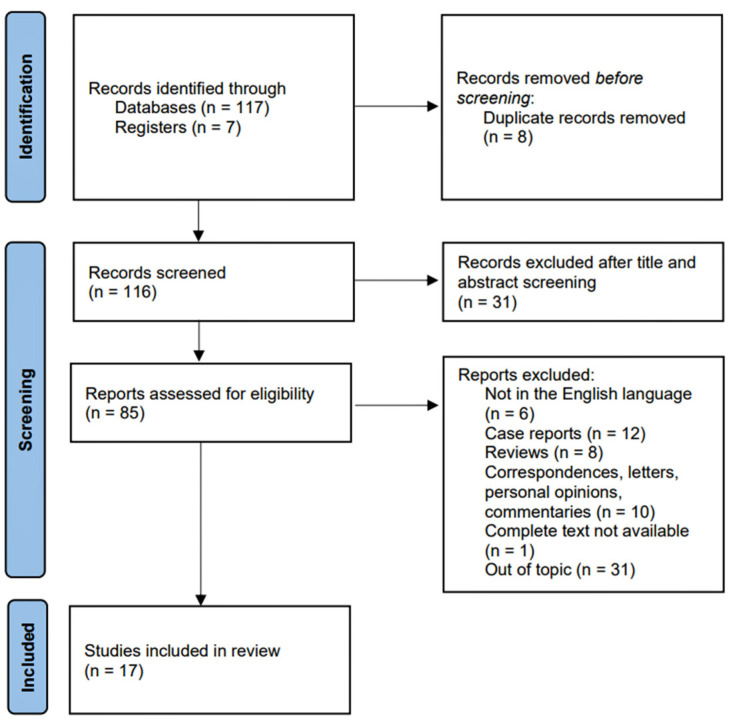
**PRISMA flow diagram for selection process.** A total of 124 studies were identified via electronic research. A total of 116 studies were eligible after duplicate removal, and 85 after title and abstract analyses. After checking all the inclusion and exclusion criteria, 17 studies were included in our review.

**Figure 2 cancers-14-05955-f002:**
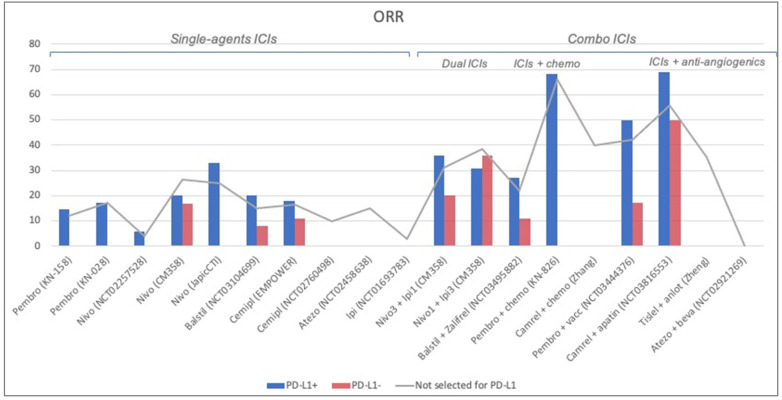
**Overall response rate (ORR) of the included studies.** ORR of the studies for patients not selected for Programmed Death Ligand 1 (PD-L1) status (*gray line*), PD-L1-positive patients (*blue bar*), and PD-L1-negative patients (*red bar*) are reported. In patients not selected for PD-L1, ORR ranged from 0% to 26.3% with single-agent ICIs, reaching 38.4% with dual ICIs association, 65.9% with the combination of ICIs and chemotherapy, 55.6% with ICIs plus tyrosine-kinase inhibitors (TKIs). In PD-L1-positive patients, ORR ranged from 5.9% to 33% with single-agent ICIs, from 27% to 36% with dual ICIs, reaching 68.1% and 69% when ICIs were combined with chemotherapy and TKIs, respectively. In PD-L1-negative women, ORR was 0–16.7% to single-agent ICIs, 11–35.8% to double ICIs, and reached 50% after ICIs + TKIs.

**Figure 3 cancers-14-05955-f003:**
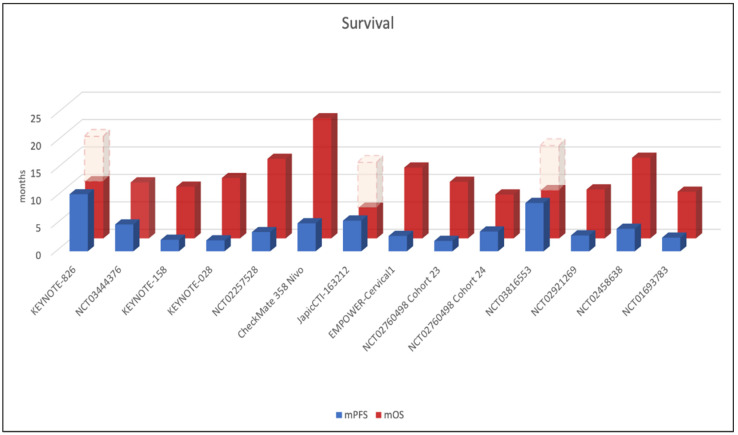
**Survival of the included studies.** Median progression-free survival (mPFS–blue bar) and overall survival (mOS–*red bar*) of the included studies are reported. A total of 13 studies reported mPFS that ranged from 2 to 10.4 months. mOS was reported by 14 studies and ranged from 8 months to not reached (*dashed lines indicate ‘not reached’ values*).

**Table 1 cancers-14-05955-t001:** Main characteristics of trials of ICIs in advanced CC.

Author	Study Name	Phase	Target Population	Administered Drugs	Primary EP	Secondary EP	PD-L1 Detection Method	Cut-Off for Positivity
Frenel et al. [24]	KEYNOTE-028 (NCT02054806)	Ib	Pre-treated PD-L1^+^ CC (*n* = 24)	Pembrolizumab 10 mg/kg q2w	ORR	Safety	22C3 (Merck)	1%
Youn et al. [25]	NCT03444376	II	Pre-treated HPV16/18^+^ CC (*n* = 36) *PD-L1^+^: n = 20* *PD-L1^−^: n = 6* *HPV16^+^: n = 20* *HPV18^+^: n = 8*	Pembrolizumab 200 mg q3w + GX-188E 2 mg (DNA vaccine)	ORR	Safety, DoR, OS, PFS6	22C3 pharmDx (Agilent), CPS	1%
Chung et al. [26]	KEYNOTE-158 (NCT02628067)	II	Pre-treated CC (*n* = 98) *PD-L1^+^: n = 82*	Pembrolizumab 200 mg q3w	ORR	Safety	22C3 pharmDx (Agilent), CPS	1%
Colombo et al. [27]	KEYNOTE-826 (NCT03635567)	III	CC (*n* = 617; 20% naïve; Pembro *n* = 307 vs. PBO *n* = 309) *PD-L1^+^ (CPS ≥ 1): n = 548* *PD-L1^+^ (CPS ≥ 10): n = 317*	Pembrolizumab 200 mg vs. PBO q3w + paclitaxel 175 mg/mq + CDDP 50 mg/mq or CBDCA 5 mg/mL/min ± bevacizumab 15 mg/kg q3w	OS PFS	DoR, ORR, 12 mos PFS rate	22C3 pharmDx (Agilent), CPS	1%
Santin et al. [28]	NCT02257528/NRG-GY002	II	Pre-treated CC (*n* = 26)	Nivolumab 3 mg/kg q2w	ORR	-	E1L3N (Cell Signaling), CPS	1%
			*PD-L1^+^: n = 17*					
Naumann et al. [29]	CheckMate 358 (NCT02488759)	I/II	Pre-treated CC (*n* = 19)	Nivolumab 240 mg q2w	ORR	DoR, OS, PFS, Safety	28-8 PharmDx (Dako), CPS	1%
			*PD-L1^+^: n = 10*					
			*PD-L1^−^: n = 6*					
Oaknin et al. [30]			Pre-treated CC N3I1 cohort *n* = 45 N1I3 cohort *n* = 112	Nivolumab 3 mg/kg q2w + Ipilimumab 1 mg/kg q6w (N3I1) Nivo1 mg/kg + Ipi 3 mg/kg q4w (x4) → Nivolumab 240 mg q2w (N1I3)				
Tamura et al. [31]	JapicCTI-163212	II	Pre-treated CC (*n* = 20)	Nivolumab 240 mg q2w	ORR	DCR, OS, PFS, DoR	28-8 PharmDx (Dako), CPS	1%
O’Malley et al. [32]	NCT03104699	II	Pre-treated CC (*n* = 161) *PD-L1^+^ (CPS ≥ 1): n = 99* *PD-L1^−^: n = 43*	Balstilimab 3 mg/kg q2w	ORR	DCR, DoR	28-8 PharmDx (Dako), CPS	1%
O’Malley et al. [33]	NCT03495882	II	Pre-treated CC (*n* = 143) *PD-L1^+^ (CPS ≥ 1): 55%* *PD-L1^−^: 25%*	Balstilimab 3 mg/kg q2w + zalifrelimab 1 mg/kg q6w	ORR	DoR, safety	28-8 PharmDx (Dako), CPS	1%
Tewari et al. [34]	EMPOWER-Cervical1/ GOG-3016 /ENGOT-CX9/NCT03257267	III	Pre-treated CC (*n* = 608)	Cemiplimab 350 mg q3w vs. single-agent chemo	OS	PFS, Safety	SP263 Ventana (Roche), TPS	1%
Rischin et al. [35]	NCT02760498	I	Pre-treated CC (*n* = 20): cohort 23: *n* = 10 cohort 24: *n* = 10	Cohort 23: Cemiplimab 3 mg/kg q2w	Safety	ORR, DCR, DoR, PFS, OS	SP263 Ventana (Roche), TPS	1%
				Cohort 24: Cemiplimab + RT				
Lan et al. [36]	NCT03816553 (CLAP)	II	Pre-treated CC (*n* = 45)	Camrelizumab 200 mg q2w + apatinib 250 mg OD	ORR	PFS, OS, DoR, Safety	28-8 PharmDx (Dako), CPS	1%
			*PD-L1^+^ (CPS ≥ 1): n = 30* *PD-L1^−^: n = 10*					
Zhang et al. [37]	/	II	Naïve CC (*n* = 35)	Camrelizumab 200 mg + NAB-paclitaxel 260 mg/mq + CBDCA AUC 5 q3w (x6) → camrelizumab 200 mg q3w	ORR	Safety	NA	NA
Zheng et al. [38]	/	II	Pre-treated CC (*n* = 25)	Tislelizumab 200 mg + anlotinib 10 mg OD d1-14 q3w	ORR	DCR, DoR, PFS, OS, Safety	NA	NA
Friedman et al. [39]	NCT02921269	II	Pre-treated CC (*n* = 11)	Atezolizumab 1200 mg + bevacizumab 15 mg/kg q3w	ORR	DCR, OS, PFS, Safety	E1L3N, CPS	1%
Tabernero et al. [40]	NCT02458638	II	Pre-treated CC (*n* = 27)	Atezolizumab 1200 mg	NPR	ORR, DoR, PFS, OS, Safety	NA	NA
Lheureux et al. [41]	NCT01693783	I/II	Pre-treated CC (*n* = 42)	Ipilimumab 10 mg/kg q3w (x4) → q12w (x4)	ORR, Safety	-	NA	NA

AUC: area under the curve; CBDCA: carboplatin; CC: cervical cancer; CDDP: cisplatin; CPS: combined positive score; DCR: disease control rate; DoR: duration of response; EP: endpoint; HPV: human papillomavirus; ITT: intention to treat; mOS: median overall survival; mPFS: median progression-free survival; NA: not available; NPR: non-progression rate; OD: once daily; ORR: objective response rate; OS: overall survival; PD-L1: programmed death ligand 1; PFS: progression-free survival; PFS6: progression-free survival at 6 months; q2/3/4/6/12w: every 2/3/4/6/12 weeks; TPS: tumor-proportional score.

**Table 2 cancers-14-05955-t002:** Results of included trials of ICIs in advanced CC.

Study Name	Administered Drugs	Nr. of Patients	Results					
			ORR	DCR	mDoR	mPFS	mOS	Safety
** *ICIs single agent* **								
KEYNOTE-028	Pembrolizumab	*n* = 24	17%	NA	5.4 mos	2 mos	11 mos	TRAEs 75% No >G3
KEYNOTE-158		*n* = 98	12.2%	30.6%	NR	2.1 mos	9.4 mos	TRAEs 65.3%, ≥G3 AEs 12.2%
		*PD-L1^+^: n = 82*	14.6%	32.9%			11 mos	
NRG-GY002	Nivolumab	*n* = 26	4%	NA	3.8 mos	3.5 mos	14.5 mos	TRAEs 84%, ≥G3 32%
		*PD-L1^+^: n = 17*	PD-L1^+^: 5.9% PD-L1^−^: 0%					
JapicCTI-163212		*n* = 20	Overall: 25%	NA	NA	5.6 mos	mOS: NR; 6 mos OS: 84%	TRAEs 65%, ≥G3 20%
			PD-L1^+^: 33%					
			PD-L1^−^: 0%					
CheckMate 358		*n* = 19	26.3%	68.4%	NR	5.1 mos	21.9 mos	TRAEs 63.2%, ≥G3 21.1%
		*PD-L1^+^: n = 10*	20%	70%				
		*PD-L1^−^: n = 6*	16.7%	50%				
NCT03104699	Balstilimab	*n* = 161	15%	Overall: 49.3%	15.4 mos	NA	NA	TRAEs 71.4%, ≥G3 11.8%
		*PD-L1: n = 99*	20%					
		*PD-L1^−^: n = 43*	7.9%					
EMPOWER-Cervical1/ GOG-3016 /ENGOT-CX9	Cemiplimab vs. single-agent chemo	*n* = 608	16.4% vs. 6.3	NA	NA	2.8 vs. 2.9 mos	12 vs. 8.5 mos	≥G3 AEs 45% vs. 53.4%
			Cemi: PD-L1^+^: 18%, PD-L1^−^: 11%				PD-L1+: 13.9 vs. 9.3 mos; PD-L1-: 7.7 vs. 6.7 mos	
NCT02760498 (Cohort 23)	Cemiplimab	*n* = 10	10%	40%	11.2 mos	1.9 mos	10.3 mos	TRAEs 90%, ≥G3 10%
NCT02458638	Atezolizumab	*n* = 27	14.8%	NA	2.99 mos-1.27 years	4.1 mos	14.7 mos	TRAEs 64.3%, ≥G3 10.7%
NCT01693783	Ipilimumab	*n* = 42	2.9%	NA	NA	2.5 mos	8.5 mos	≥G3 TRAEs 9.5%
** *Double ICIs (anti-PD1 + anti-CTLA4)* **								
CheckMate 358	Nivolumab + ipilimumab (N3I1)	*n* = 45	Overall: 31.1% PD-L1^+^: 36% PD-L1^−^: 20%	NA	NA	NA	NA	N1I3 hepatitis 16%
	Nivolumab + ipilimumab (N1I3)	*n* = 112	Overall: 38.4% PD-L1^+^: 35.8% PD-L1^−^: 30.6%					
NCT03495882	Balstilimab + zalifrelimab	*n* = 143	22%	NA	NR	NA	NA	35% irAEs, ≥G3 irAEs 10.5%, 2 deaths
		*PD-L1^+^: 55%*	27%					
		*PD-L1^−^: 25%*	11%					
** *ICIs + chemotherapy* **								
KEYNOTE-826	Pembrolizumab vs. PBO + paclitaxel + CDDP/CBDCA ± bevacizumab	*n* = 617 (20% naïve): Pembro *n* = 307 vs. PBO *n* = 309	65.9% vs. 50.8%	NA	18.0 vs. 10.4 mos	10.4 vs. 8.2 mos	mOS: NR; 24-mos OS rate: 50.4% vs. 40.4%	Pembro arm: irAEs 33.9%, ≥G3 AEs 81.8%, 14 deaths PBO arm: irAEs 15.2%, ≥G3 AEs 75.1%, 14 deaths
		*PD-L1^+^ (CPS ≥ 1): n = 548*	68.1% vs. 50.2%		18.0 vs. 10.4 mos	10.4 vs. 8.2 mos	24-mos OS rate: 53% vs. 41.7%	
		*PD-L1^+^ (CPS ≥ 10): n = 317*	69.6% vs. 49.1%		21.1 vs. 9.4 mos	10.4 vs. 8.1 mos	24-mos OS rate: 54.4% vs. 44.6%	
/	Camrelizumab + NAB-paclitaxel + CBDCA	*n* = 35	40%	92%	NA	NA	NA	RCCEP 65.7% ≥G3 20%
** *ICIs + anti-angiogenics* **								
NCT03816553	Camrelizumab + apatinib	*n* = 45	55.6%	NA	NR	8.8 mos	NR	TRAEs 95.6%, ≥G3 71.1%, irAEs 33.3%
		*PD-L1^+^: n = 30*	69%			NR		
		*PD-L1^−^: n = 10*	50%			5.2 mos		
/	Tislelizumab + anlotinib	*n* = 25	35.3%	94.1%	NA	NR	NR	TRAEs 100% ≥G3 0.06%
NCT02921269	Atezolizumab + bevacizumab	*n* = 11	0%	60%	NA	2.9 mos	8.9 mos	TRAEs 72%, ≥G3 36.4%
** *Other combinations* **								
NCT03444376	Pembrolizumab + GX-188E (DNA vaccine)	*n* = 36	42%	58%	3.3–13.6 mos	4.9 mos	10.2 mos	TRAEs 44%, ≥G3 11%
		*PD-L1^+^: n = 20*	50%	65%				
		*PD-L1^−^: n = 6*	17%	29%				
		*HPV16^+^: n = 20*	45%	60%				
		*HPV18^+^: n = 8*	33%	50%				
NCT02760498 (Cohort 24)	Cemiplimab + RT	*n* = 10	10%	60%	6.4 mos	3.6 mos	8 mos	TRAEs 100% ≥G3 30%

AE(s): adverse event(s); AUC: area under the curve; CBDCA: caroplatin; CC: cervical cancer; CDDP: cisplatin; CI: confidence interval; CPS: combined positive score; CR: complete response; DCR: disease control rate; G3: grade 3; HPV: human papillomavirus; irAE(s): immune-related adverse event(s); mDOR: median duration of response; mos: months; mOS: median overall survival; mPFS: median progression-free survival; NA: not available; NR: not reached; ORR: objective response rate; OS: overall survival; PD-L1: programmed death ligand 1; PFS: progression-free survival; PR: partial response; RCCEP: reactive cutaneous capillary endothelial proliferation; SD: stable disease; TRAE(s): treatment-related adverse event(s).

**Table 3 cancers-14-05955-t003:** Ongoing trials on ICIs and combinations in advanced/recurrent CC.

Clinicaltrials.gov Reg. Number	Phase	ICI	Combination
NCT04641728	II	Pembrolizumab	Olaparib
NCT04865887	II	Pembrolizumab	Lenvatinib
NCT03367871	II	Pembrolizumab	CTX, Paclitaxel, Bevacizumab
NCT04230954	II	Pembrolizumab	Cabozantinib
NCT02635360	II	Pembrolizumab	CTX, RT
NCT04483544	II	Pembrolizumab	Olaparib
NCT03786081	I-II	Pembrolizumab	Tisotumab Vedotin, Carboplatin, Bevacizumab
NCT03108495	II	Pembrolizumab	LN-145 (autologous TILs)
NCT04652076	I-II	Pembrolizumab	NP137 (anti-Netrin1 Ab), Paclitaxel, Carboplatin
NCT05082259 (ASTEROID)	I	Pembrolizumab	ASTX660
NCT03476681	I-II	Pembrolizumab	NEO-201 (Ab)
NCT03236935	Ib	Pembrolizumab	L-NMMA
NCT04651127	I-II	Pembrolizumab	Toripalimab, Chidamide
NCT04301011	I-II	Pembrolizumab	TBio-6517
NCT03849469	I	Pembrolizumab	XmAb^®^22841
NCT04895709	I-II	Nivolumab	-
NCT02379520	I	Nivolumab	HPV-Specific T Cells, Cytoxan, Fludarabine
NCT04646005	II	Cemiplimab	ISA101b Vaccine
NCT01693783	II	Ipilimumab	-
NCT03752398	I	Ipilimumab	XmAb^®^23104
NCT03826589	NA	Avelumab	Axitinib
NCT03260023	I-II	Avelumab	TG4001
NCT04300647	II	Atezolizumab	Tiragolumab
NCT03614949	II	Atezolizumab	RT
NCT03340376	II	Atezolizumab	Doxorubicin
NCT04405349	II	Atezolizumab	VB10.16
NCT03738228	I	Atezolizumab	Cisplatin, RT
NCT03556839 (BEATcc)	III	Atezolizumab	Bevacizumab, Cisplatin, Carboplatin, Paclitaxel
NCT04405349	II	Atezolizumab	VB10.16 Vaccine
NCT03073525	II	Atezolizumab	Vigil
NCT04800978	II	Durvalumab	BAVC-C Vaccine
NCT03277482	I	Durvalumab, Tremelimumab	RT
NCT03452332	I	Durvalumab, Tremelimumab	RT
NCT04918628	II	Durvalumab	Sintilimab
NCT03439085	II	Durvalumab	Vaccine MEDI0457
NCT04646005	II	Cemiplimab	ISA101b
NCT04068753	II	Dostarlimab	Niraparib

Ab: antibody; CC: cervical cancer; CTX: chemotherapy; HPV: human papillomavirus; ICI: immune checkpoint inhibitor; L-NMMA: NG-monomethyl-L-arginine; RT: radiotherapy; TILs: tumor-infiltrating lymphocytes.

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
