# Peer review of "Beyond Platinum, ICIs in Metastatic Cervical Cancer: A Systematic Review"

_cancers, 2022, doi:10.3390/cancers14235955_

Round 1
Reviewer 1 Report
The authors reported “Beyond platinum: ICIs in metastatic cervical cancer. A systematic review”.
This is interesting observation.
Unfortunately, this paper is only a list of results. As a systematic review, no comparisons were made regarding patient backgrounds and outcomes of the articles. The advantages of ICI therapy for cervical cancer dominate half of the discussion, and there is no discussion of the results for this review. I cannot recommend publication of this manuscript.
Author Response
We thank the reviewer for the time dedicated to our work. We systematically reviewed all the available information regarding clinical trials using ICIs in the metastatic setting of CC, focusing on the main efficacy and safety points, and also mentioning biomarkers that have been investigated so far. This work was conducted following the PRISMA 2020 guidelines for systematic reviews. However, we updated the manuscript with more discussion elements and ordered the results in order to make the paper more suitable.
Reviewer 2 Report
Congratulations to the authors. This paper is very well designed in terms of statistical soundness, and provides a comprehensive, critically assessed overview of the evolving field of immune checkpoint inhibitors in advanced CC.
Some minor comments:
Introduction: The introduction provides the needed background on CC treatment, however, is difficult to follow as a single paragraph. Would recommend breaking up paragraphs by major topics or treatment modalities, and the last paragraph as an introduction to what will be covered in the review.
In the 'characteristics of included studies' section, the authors provide an explanation of the study end-points. Please include a sentence explaining the disease control rate.
Table 1 could be streamlined/organized to be more easily digestible by the reader perhaps by grouping or simplifying what is presented. For example, Keynote 826 could be broken into two sub-boxes for CPS>1 vs. 10 or otherwise additional organizing. Most are more straightforward and don't require additional breaks.
Figure 2 is difficult to understand. The representation of inclusion of PD-L+ vs PD-L1- patients is very helpful, however, some improvements could be made. At a minimum, the authors should indicate on the chart which studies used combination ICI vs. single agent vs. combination with another non ICI drug. Consider grouping the bars by agent used rather than study name.
Please explain what G3 AEs mean (grade 3). Include a reference to CTCAE.
Section 3.2.1. Pembrolizumab - you are describing a study that only enrolled HPV 16/18+ patients. Please also include how their HPV status was determined.
It would be useful to include a section/indicate in the text if all studies used a comparable cut-off to determine PD-L1 positivity (CPS? TPS? >1?). The authors are inconsistent in reporting the PD-L1 evaluation method for these studies.
The authors should comment on improved ORR in the single agent vs. combination ICI setting. A table comparing single agent Nivo vs. Ipi/Nivo for key endpoints would be helpful (ORR, PFS, OS, AEs), potentially for other single agent/combinations.
The sections on Pembro and Nivo could and should be addressed in paragraphs.
The first part of the discussion (focusing on immune cell roles in CC relative to HPV infection) would be better suited in the introduction section and indeed repeats some of the information in the introduction but could be restated as it relates to the studies or findings only in the final sections of the discussion.
Again, the discussion needs to be broken into topical paragraphs, and bring in the information presented in the results section and applied to clinical trials more effectively.
Reviewer 3 Report
The Authors described an interesting and exhaustive systematic review on one of the hottest topic of the moment, that is the role of ICI in patients with cervical cancer. The paper is well written and easily to read, complete in each part.
I have only minor comments:
- Introduction: please correct ICI with ICIs
- The text misses authors' contribution paragraph
Author Response
We are extremely grateful to the reviewer for the comments. We modified the manuscript following the suggestions: we corrected ICIs in the Introduction section, and added the authors’ contribution paragraph.
Reviewer 4 Report
In the manuscript: “Beyond platinum: ICIs in metastatic cervical cancer. A systematic review” the authors discuss that Immune-checkpoint inhibitors (ICIs) might represent an appealing strategy for advanced cervical cancer (CC). An interesting topic that contributes to knowledge in the area, but certain issues must be corrected.
Major review
1. In the abstract and introduction, the gap of the manuscript must be mentioned, mentioning what is the gap that the manuscript will fill within the current knowledge, and the topics that will be reviewed in the manuscript.
2. In Figure 1, please, add a description of the diagram.
3. The results should be improved by adding an explanation at the beginning of their description, mentioning why the ICI in question was reviewed and what is its importance for CC.
4. Please conclude each section of the review. That is, “in conclusion, these data demonstrated that” or “together this data shows…”, especially for each ICI.
5. The discussion part has a lot of information that could go in the introduction, it seems to me that the authors could enrich the introduction with this information. In the discussion, the authors can focus on the results obtained and contrast them with the works already published in the literature.
6. The systematic review does not contain any association between HPV, CC, and ICIs, it was expected that this analysis would be done since the function of HPV is crucial for the development and treatment of CC, even the authors mention it both in their introduction and in their discussion. This analysis would greatly enrich the work of the systematic review, so the authors are encouraged to do so.
Minor reviews
1. Improve figure 1. It looks pixelated.
2. Author contribution and supplementary material sections are not defined
Round 2
Reviewer 1 Report
The authors reported “Beyond platinum: ICIs in metastatic cervical cancer. A systematic review”. Compared to the previous paper, the discussion has been extensively revised and rewritten. As a result, it is easier to understand. This paper deserves a publication.
Author Response
We would like to thank again the reviewer for the valuable comment.
Reviewer 4 Report
Minor reviews
The authors did not provide a response letter.
The authors still do not provide the gap of their article, they do not mention what area they cover that other review articles have not.
Author Response
We would like to thank the reviewer for the valuable comments.
As requested, we better specified the gap of knowledge filled by our work in the 1.3 paragraph, and provided a response letter.